# Effects of an 11-Week Detraining, Imposed by the COVID-19 Confinement, on Handball Players’ Shoulder Rotator Isokinetic Profile, Shoulder Range of Motion, and Ball Release Velocity

**DOI:** 10.3390/medicina59091548

**Published:** 2023-08-25

**Authors:** Nuno Batalha, João Paulo Sousa, Orlando Fernandes, Eduardo Dias, Jose A. Parraca, Santos Villafaina

**Affiliations:** 1Departamento de Desporto e Saúde, Escola de Saúde e Desenvolvimento Humano, Universidade de Évora, 7000-727 Évora, Portugal; jsousa@uevora.pt (J.P.S.); orlandoj@uevora.pt (O.F.); eduardo_dias_9@hotmail.com (E.D.); jparraca@uevora.pt (J.A.P.); svillafaina@unex.es (S.V.); 2Comprehensive Health Research Centre (CHRC), University of Évora, 7000-727 Évora, Portugal; 3Grupo de Investigación Actividad Física y Calidad de Vida (AFYCAV), Facultad de Ciencias del Deporte, Universidad de Extremadura, 10003 Cáceres, Spain

**Keywords:** handball, detraining, shoulder, range of motion, strength

## Abstract

*Background and Objectives:* The COVID-19 confinement significantly impacted the physical condition of athletes. However, the detraining impacts of this period on the shoulder rotator and range of motion in handball players have not been studied. Thus, the main aim of this study was to investigate the effect of this 11-week detraining period, imposed by the COVID-19 pandemic confinement, on the shoulder rotator isokinetic profile (peak torque, ratio, fatigue index), shoulder rotator and flexion range of motion, and ball release velocity in handball players. *Materials and Methods:* A total of 16 handball players, with a mean age of 22.38 (5.28) years, participated in this study. The isokinetic strength was assessed using two protocols (three repetitions at an angular velocity of 60°/s and 20 repetitions at an angular velocity of 180°/s). In addition, the range of motion and ball release (at jump and standing shots) were measured. All these measurements were assessed before and after the COVID-19 confinement. *Results:* The results showed a significant reduction in the peak torque of the external rotation of their dominant and non-dominant shoulders. In addition, confinement significantly increased the fatigue index of external rotation and internal rotation and reduced the range of motion of internal rotation. Additionally, the ball release velocity during standing and jump shots was significantly reduced. *Conclusions:* These results suggested that strengthening external and internal rotation as well as recovering the internal rotation range of motion may be necessary after a detraining period in order to prevent shoulder injuries.

## 1. Introduction

Currently, handball is played in more than 199 countries [1], with around 27 million practitioners worldwide. It is one of the most popular sports in Europe [2]. This modality is considered a collective, complex, and explosive sport [3] that is characterized by ball throwing, change of directions, jumps or repetitive sprints [2], changes in directions, passing, grabbing, and a high pace of offensive and defensive action during the competitive period [4]. Handball practice has several health benefits, such as an improvement in cardiovascular, metabolic, muscular, and psychosocial capacity [5].

The high ball velocity achieved in throwing has emerged as one of the most essential performance factors in handball [6]. Previous studies have investigated the role of strength and power in the upper and lower extremities in achieving this high throwing ball velocity [7,8]. Furthermore, ball acceleration is primarily a result of shoulder internal rotation and elbow extension [9].

Due to repetitive throwing tasks and blocks to the upper limb, handball players’ shoulders are vulnerable to acute and overuse injuries. Overall, the prevalence of shoulder injuries ranges between 17% to 41% in handball players [10,11], with overuse conditions dominating [12,13]. Previous studies have considered that shoulder internal rotation (IR) and external rotation (ER) weakness, as well as range of motion (ROM), are risk factors for shoulder disorders in handball players [10,14]. Consequently, in order to obtain the profile of shoulder IR and ER, the isokinetic strength has been measured [15,16] in order to establish correlations with the achieved ball velocity [17] and strength field test performance in female handball players [18]. In this regard, the isokinetic torque profile is relevant in this sport, since it provides an objective and quantitative assessment of shoulder-joint-specific muscle performance and therefore of strength. The isokinetic dynamometer applied a force throughout a specified range of motion, providing relevant information about dynamic concentric and/or eccentric muscle contractions [19]. It is considered as the gold standard for measuring muscle strength and muscle endurance through a specified range of movement [20]. Thus, previous studies have used this technique to evaluate the potential risk of injury or the effect of training programs in the shoulder joint [21]. In the shoulder muscle in particular, this assessment could help to identify weaknesses and imbalances [22].

In addition, handball is characterized by repetitive actions (such as passing or shooting) performed with the throwing arm. This could lead to inter-limb asymmetry between the throwing and non-throwing arm. Asymmetries in handball players between dominant and non-dominant sides have been studied for upper and lower limbs [23,24]. However, as commented before, the prevalence of shoulder injuries reached a high prevalence [10,11]. In this regard, previous studies have identified that asymmetries between dominant and non-dominant shoulders in young handball players could increase the risk of injury [25]. In addition, these repetitive movements generate neuromuscular fatigue and, therefore, changes in the technique that could lead to injuries [26,27]. In this line, a previous study reported that shoulder external rotation muscle fatigue contributed to altered scapular muscle activation and kinematics, predisposing the shoulder to injury [27]. Therefore, this study will be focused on differences between dominant and non-dominant upper limbs.

The COVID-19 pandemic suspended all sports training and competitions, forcing people to remain confined to their homes. According to Hermassi, et al. [28], during COVID-19 confinement, handball players significantly reduced their physical activity and increased their sitting time. Consequently, some studies have investigated the effect of home-based exercise training on the maintenance of physical condition during confinement [29]. Font, et al. [30] demonstrated that a 9-week structured home-based intervention during the COVID-19 lockdown maintained lower limb explosive strength but was insufficient to preserve aerobic capacity in elite handball players. Despite a home-based strength and endurance program, Fikenzer, Fikenzer, Laufs, Falz, Pietrek and Hepp [29] obtained similar results, reporting a reduction in the endurance capacity of elite handball players who did not participate in team training. However, to the best of our knowledge, there are no published data on the isokinetic profile of shoulder rotators, likely due to the complexity in the measurement data of shoulder strength before and after the COVID-19 pandemic. Since detraining effects on these muscles may shed light on how to arrange training throughout various seasons, such as preseason or post-injury, this information would be useful to coaches and exercise specialists.

Therefore, the purpose of this study was to investigate the effect of an 11-week detraining period, imposed by the COVID-19 confinement, on the shoulder rotator isokinetic profile (peak torque, ratio, fatigue index), shoulder rotator and flexion ROM, and ball release velocity (BRV) in handball players. We hypothesized that the shoulder rotator peak torque (PT) and ratio, shoulder range of motion (ROM), and BRV would decrease following the COVID-19 confinement, while the ER and IR fatigue indices would rise.

## 2. Materials and Methods

### 2.1. Participants

A total of 16 handball players participated in this study, with a mean playing experience of 12 (2.5) years. All participants competed in the 2nd national Portuguese division and had a training volume of four sessions (90 min) per week. The sample mean age was 22.38 (5.28) years old with 1.77 (0.38) meters of height, 83.44 (15.60) kg of weight, and a resulting mean body mass index (BMI) of 26.32 (4.67) kg/m^2^. Two of the players were left-handed. Participants were excluded if: (a) they had an injury to the shoulder at the time of the study; (b) they had a history of injury to the shoulder in the last six months; (c) they had another type of injury that prevented them from practicing sports; or (d) they participated in another shoulder injury prevention program regularly.

A post-hoc power analysis performed with the G*Power software (v.3.1.9.7; Kiel University, Kiel, Germany) estimated a power of 0.98 to detect differences in the IR and 0.39 for ER, with an error probability of 0.05 with a total sample size of 16.

According to the Declaration of Helsinki, all participants were informed about the research procedures and gave written informed consent. The procedures were approved by the bioethical committee of the host University (N20081).

### 2.2. Procedures

Participants were recruited from a local handball club. The initial objective was the implementation and evaluation of a handball-player shoulder injury prevention intervention. Initial assessments were carried out between 18 January and 21 January 2021. However, the Portuguese government imposed home confinement owing to the COVID-19 pandemic, which undermined this goal. Initial assessments were made at that time, but the intervention had not yet started. The confinement period lasted 10 weeks, between 2 February and 12 April. The week after the end of the confinement period (April 2021), all players underwent the same evaluation procedures once more. Therefore, a total of 11 weeks of detraining were performed. During the confinement, all handball players were advised to follow a training program at home that included the following: 45 min of aerobic training (cycling or running, if possible) and 30 min of strength training (core training, arm and squats). It was agreed that the plan would be carried out three times a week.

### 2.3. Instruments and Outcomes

#### 2.3.1. Isokinetic Profile

The shoulder rotator isokinetic profile was determined using an isokinetic dynamometer (Biodex System 3–Biodex Corp., Shirley, NY, USA). A neutral shoulder position, with the arm at 90° of abduction and elbow flexion, was defined as the initial test position. All the participants performed a ROM from neutral rotation to 90° of external rotation. Recommendations were considered for the correction of the gravitational effect. The upper limb was relaxed upper in the neutral starting position, and the glenohumeral joint center aligned with the dynamometer axis (see Figure 1).

Regarding the angular velocity tests performed on the isokinetic dynamometer, considering that most studies carried out with Overhead Athletes use at least two different execution velocities, the most common of which is 60°/s [21], we chose the use of the following protocols:Performing three repetitions at an angular velocity of 60°/s.Executing 20 repetitions at an angular velocity of 180°/s.

Throughout both protocols, the evaluator provided continuous verbal encouragement. Prior to the test, each participant was informed on the testing procedure, with an emphasis on exerting maximum effort within their tolerance level. To familiarize participants with movement, they performed three repetitions at each speed. For each shoulder, the 60°/s speed protocol was performed first, followed by the 180°/s speed protocol. The protocols were separated by a two-minute interval of rest. The order of dominant and non-dominant shoulders was determined arbitrarily.

Peak torque was used to monitor the strength of the shoulder rotators, which can be defined as the maximum torque produced at any point in the ROM [31]. In addition, the ER/IR ratio [the ratio between ER and IR PT values: (ER − PT/IR − PT) × 100] was calculated to evaluate the rotator strength balance. The Fatigue Index (FI) was defined in accordance with the Biodex recommendations, as follows: Fatigue Index = [(W1 − W2)/W1] × 100, where W1 and W2 represent the mechanical work of the first third (first four trials) and last third (last four trials) of the repetitions, respectively. This variable was only extracted using the 20 repetitions at 180°/s protocol as a measure of muscle fatigue.

#### 2.3.2. Shoulder Range of Motion

External and internal shoulder rotator ROM was evaluated passively with a manual goniometer, with the participant supine on a stretcher, the shoulder abducted to 90° and the test side humerus supported on the evaluation surface. With the forearm in neutral position, the goniometer axis was aligned with the olecranon process of the ulna. The stationary arm was perpendicular to the floor and the movement arm in line with the ulnar side of the forearm from the axis point to the ulnar styloid process. For shoulder flexion, also in supine position, the axis of the goniometer was placed laterally to the middle of the humeral head. The stationary arm was parallel to the trunk, while the movement arm was in line with the humerus lateral epicondyle (Figure 2).

Participants were encouraged to progress slowly into ER, IR and flexion, so that it was possible to follow the movement with the goniometer.

#### 2.3.3. Ball Release Velocity

The participants completed jump throws from 9 m after running steps and 6-m line throws after running steps without the jump. The ball velocity was measured using a High-Speed Camera (Casio Exilim EX-FH25) at a 210 Hz frequency. The camera was set up on a fixed tripod on level ground, perpendicular to the subject’s movement in the sagittal plane, and as far away from the action as possible to reduce perspective error. The admissible error will be inside 0.0047619 of a second. The intra-operator consistency between digitalizations was significant at *p* < 0.001 with a mean Pearson’s correlation coefficient value of 0.962 ± 0.0271. The calibration was carried out in accordance with the recommendations of the International Society of Biomechanics, and we employed the direct linear transformation (DLT). Despite knowing that the minimal number of points required to estimate the 2D-DLT parameters is four, we collected seven non-collinear points utilizing a PIPE (2.60 m high) stretched over four meters, forming a plane with dimensions of 4 m × 2.60 m. The frames were digitalized using the Tracker© 5.15 software. The ball velocity was calculated using eight frames, four images before the ball left the player’s hand and four images immediately after the ball left the player’s hand. Using the projectile motion equations, the horizontal movements had an equation (polynomial of degree 1), while the vertical movements had a quadratic equation (polynomial of degree 2).

### 2.4. Statistical Analysis

Statistical analyses were conducted using the International Business Machines SPSS Statistics Corp., Armonk, NY, USA, version 25. According to the Shapiro−Wilk test results, parametric and non-parametric analyses were conducted when appropriate. Regarding isokinetic profile variables, a ROM and BRV Wilcoxon signed-rank test and Paired Sample *t*-test were conducted between pre- and post-evaluations. The effect sizes, Cohen’s *d* for parametric tests and r for non-parametric tests were calculated for each comparison as well as classified using the following classification: (a) for non-parametric tests, <0.1 as a small effect, between 0.1 and 0.5 as a medium effect, and >0.5 as a large effect; and (b) for parametric tests, <0.2 as a small effect, between 0.2 and 0.8 as a medium effect, and >0.8 as a large effect [32].

## 3. Results

Table 1 shows the impact of the detraining due to confinement on isokinetic strength at 60° degrees/second. Participants exhibited a reduction in the PT of the ER in their dominant (*p* = 0.001) and non-dominant shoulders (*p* = 0.008) and in the IR of their non-dominant shoulder (*p* < 0.001). Although non-significant, the PT of the IR in their dominant shoulder (*p* = 0.052) and the ER/IR ratio of their dominant shoulder (*p* = 0.070) reached a large effect size.

Table 2 displays the effect of the confinement-induced detraining on isokinetic strength at 180 degrees/second. A significant reduction in the PT was found in the ER of the dominant (*p* < 0.001) and non-dominant shoulder (*p* = 0.002) as well as in the FI of the ER (*p* < 0.001) and IR (*p* = 0.001) of the dominant shoulder.

Table 3 shows the impact of the detraining due to confinement on shoulder ROM and BRV. The results showed that the ROM of the IR in the dominant shoulder was significantly reduced (*p* < 0.001) after confinement. In addition, the results showed a significant reduction in the BRV at jump (*p* = 0.011) and low/set shots (*p* = 0.017).

## 4. Discussion

In this study, we sought to determine how a detraining period brought on by COVID-19 pandemic confinement affected handball players’ shoulder rotator isokinetic profile, shoulder rotator range of motion, as well as BRV. The results showed a significant reduction in the PT of the ER in their dominant and non-dominant shoulders and in the IR of their non-dominant shoulder at 60°/s. Regarding 180°/s, a significant reduction in the PT was found in the ER of their dominant and non-dominant shoulder, as well as a significant increase in the FI of the ER and IR of their dominant shoulder. ROM was significantly reduced in the IR of their dominant shoulder, and BRV was also significantly reduced.

The COVID-19 pandemic has had a substantial impact on athletes’ physical and emotional health [33]. Previous research on handball players found that they spent more time sitting, while engaging in less physical exercise [28]. To prevent significant deterioration in physical condition, home-based exercises were performed [29]. However, the results showed that they were inefficient, specifically in maintaining aerobic capacity. Nevertheless, Font, Irurtia, Gutierrez, Salas, Vila and Carmona [30] showed that nine weeks of home-based exercise preserved lower-limb explosive strength. However, little is known about shoulder strength during COVID-19 confinement. Our results showed that PT of the IR and ER were reduced and that FI was increased. This is in line with previous research [34], which showed that 12 weeks of detraining produced mean torque decreases in ER and IR PT at 60°/s and 120°/s. Nevertheless, handball players were encouraged to conduct some exercises during the confinement. These exercises consisted of 45 min of aerobic training (cycling or running, if possible) and 30 min of strength training (core training, arm and squats) three times per week. These recommendations, if fulfilled by the players, were not enough to ameliorate the detraining impact on the shoulder joint. The main reason might be that prescribed exercises were not focused on external and internal rotation. In this regard, the shoulder external rotation primarily involves muscles such as the infraspinatus and teres minor, while internal rotation involves the subscapularis muscle. During the confinement (detraining) weeks, these muscles may have experienced varying degrees of atrophy, decreased neuromuscular coordination, and reduced strength and endurance. For instance, Mujika & Padilla [35] reported that at the muscle level, capillary density and oxidative enzyme activities are reduced after exercise cessation. Furthermore, even changes in the autonomic modulation, assessed by heart rate variability, have been observed in swimmers after a 5-week training cessation [36]. These decreases could explain the yield loss observed in the results.

The main decreases in PT were observed in the ER of the dominant and non-dominant shoulder at both 60°/s and 180°/s, probably because of detraining. In contrast, Fieseler, et al. [37] observed that handball players, during a playing season, significantly improved the isometric strength in ER. Regarding the ROM, a significant reduction was only found in the IR of the dominant shoulder. This is important, since increased ER and decreased IR are generally observed in overhead athletes [14,38,39]. Reduced shoulder IR can be derived from a glenohumeral IR deficit, increasing the risk of shoulder injury [40,41]. In our study, handball players after confinement exhibited a mean loss of 10.44° in the IR. Considering these results, handball players, after confinement or detraining periods, would need specific training to increase IR ROM. In this regard, previous studies have conducted different protocols that may be helpful for improving IR ROM in overhead throwing athletes [42,43]. In addition, previous studies also showed that self-regulated training and non-linear pedagogy in handball can improve running speed and jump height as well as motor skill [44,45].

Irrespective of the throwing technique, throwing velocity is a crucial performance factor in handball, and greater shooting velocities are displayed by highly performing players [6]. Previous studies suggested that the ball’s acceleration depended on the shoulder’s IR angular velocity and elbow’s extension [9,43]. Thus, a decrease in PT (above all the IR) could affect players’ performance. As commented above, although non-significant, the PT of IR was reduced (with a large effect size) after confinement. In addition, the IR fatigue index of the dominant shoulder at 180°/s was significantly increased after confinement. These findings can be related to the loss in BRV observed in both types of shots (standing and jump shots).

Repetitive movements, such as shooting in handball, generate neuromuscular fatigue and, therefore, muscular and kinematic adaptations [46] that can lead to injuries [47]. Thus, detecting shoulder fatigue is crucial to avoid injuries [48]. In our study, we conducted a fatigue protocol consisting of 20 repetitions at 180°/s. The results showed that the FI for ER and ER was significantly increased after confinement in the dominant shoulder. The non-dominant shoulder, however, did not exhibit these important modifications. Considering that shots are predominantly performed with the dominant shoulder, the risk of injury after detraining/confinement was increased. Therefore, one would be advised to carry out interventions meant to lessen ER and ER fatigue following periods of inactivity. In this regard, a previous study has explored the effects of exercise intervention after a detraining period in soccer players. Clemente, et al. [49] showed that small-side games and high-intensity interval training interventions improved physical fitness outcomes after a period of detraining; they were not able to effectively restore body composition, flexibility or lower-limb strength compared with the baseline assessments (before detraining).

This article has some limitations that should be acknowledged. Firstly, the relatively small sample size could mean that only greater differences reached the significance level. Secondly, participants were recruited from a local handball team. Therefore, a comparison with elite handball players should be made with caution. Thirdly, the results correspond to the pre- and post-COVID-19 confinement detraining period. As all players and coaches were confined and unable to leave their homes, an effective control of physical activity levels and eating habits was impossible to achieve. However, a systematic assessment via a questionnaire, which could have been useful to control the training, was not conducted.

## 5. Conclusions

The detraining period imposed by the COVID-19 confinement led to a decrease in the PT of ER in dominant and non-dominant shoulders. In addition, FI of ER and IR were significantly increased, and the IR ROM was reduced after confinement, which could increase the risk of a shoulder injury. Consequently, BRV in standing and jump shots was also reduced. As a result, after detraining phases, specific training aimed at strengthening ER and IR and recovering IR ROM is required to prevent the onset of shoulder problems. Therefore, training should focus on muscles involved in ER and IR, such as the infraspinatus and teres minor for ER and subscapularis for IR, as well as other shoulder secondary muscles and the upper arm, which contribute to a lesser extent to external and internal rotation movements. Some of these muscles include the deltoids (anterior and posterior parts), pectoralis major, and latissimus dorsi. Therefore, maintaining strength and flexibility in the IR and ER muscles is essential for optimal shoulder health and function.

## Figures and Tables

**Figure 1 medicina-59-01548-f001:**
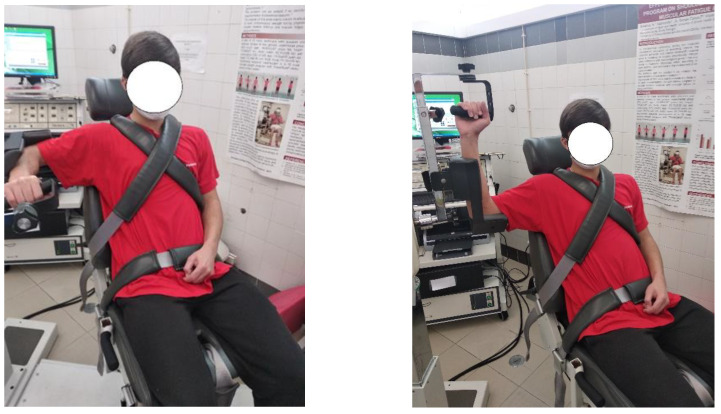
Initial and final positions during the isokinetic test.

**Figure 2 medicina-59-01548-f002:**
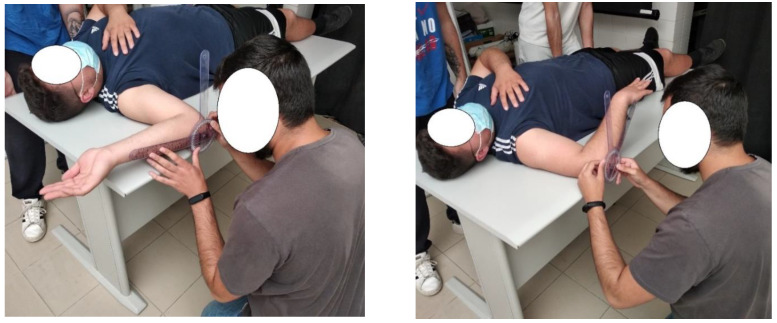
Evaluation of external and internal shoulder rotation.

**Table 1 medicina-59-01548-t001:** Effect of confinement on IR and ER PT and ER/IR ratios of the dominant and non-dominant shoulders at 60°/s.

Variable	Pre-Confinement Mean (SD)	Post-Confinement Mean (SD)	Contrast ^a^	*p*	Effect Size
Dominant shoulder at 60°/s					
ER (Nm)	41.29 (6.97)	34.90 (5.93)	4.156	0.001 *	1.039 (large)
IR (Nm)	51.18 (11.59)	47.42 (8.64)	2.110	0.052	0.528 (medium)
ER/IR ratio (%)	81.89 (10.14)	74.70 (11.30)	1.810	0.070 *^a^	0.670 (large)
Non-dominant shoulder at 60°/s					
ER (Nm)	37.53 (6.21)	33.70 (6.55)	−2.638	0.008 *^a^	0.600 (large)
IR (Nm)	47.91 (6.89)	43.17 (7.16)	7.091	<0.001 *	1.773 (large)
ER/IR ratio (%)	78.48 (7.75)	78.55 (10.84)	−0.027	0.978	0.007 (small)

*: *p* < 0.05; ^a^: *p* obtained from Wilcoxon Signed-Rank test; ER—External rotation; IR—Internal rotation.

**Table 2 medicina-59-01548-t002:** Effect of confinement on IR and ER PT and ER/IR ratios of the dominant and non-dominant shoulders at 180°/s.

Variable	Pre-Confinement Mean (SD)	Post-Confinement Mean (SD)	Contrast ^a^	*p*	Effect Size
Dominant shoulder at 180°/s					
ER (Nm)	38.39 (6.98)	33.96 (6.67)	5.554	<0.001 *	1.389 (large)
IR (Nm)	48.07 (11.60)	46.17 (9.38)	0.838	0.412	0.210 (medium)
ER/IR ratio (%)	81.37 (8.94)	74.87 (12.72)	1.658	0.118	0.415 (medium)
ER fatigue index (%)	21.95 (8.40)	28.13 (6.50)	−4.443	<0.001 *	1.111 (large)
IR fatigue index (%)	18.13 (8.28)	26.47 (9.73)	−3.464	0.001 *^a^	0.923 (large)
Non-dominant shoulder at 180°/s					
ER (Nm)	35.62 (6.83)	32.16 (6.86)	3.637	0.002 *	0.909 (large)
IR (Nm)	43.96 (7.60)	42.92 (9.60)	0.720	0.483	0.180 (small)
ER/IR ratio (%)	81.27 (9.66)	76.23 (11.79)	1.339	0.201	0.335 (medium)
ER fatigue index (%)	26.09 (10.40)	27.03 (8.39)	−0.659	0.520	0.165 (small)
IR fatigue index (%)	21.44 (7.97)	23.56 (9.10)	−1.008	0.329	0.252 (medium)

*: *p* < 0.05; ^a^: *p* obtained from Wilcoxon Signed-Rank test; ER—External rotation; IR—Internal rotation.

**Table 3 medicina-59-01548-t003:** Shoulder’s range of motion and ball release velocity.

Variable	Pre-Confinement Mean (SD)	Post-Confinement Mean (SD)	Contrast ^a^	*p*	Effect Size
Dominant shoulder					
IR (degrees)	62.94 (9.12)	52.50 (9.82)	3.314	0.001 *^a^	1.101 (large)
ER (degrees)	77.81 (13.25)	74.56 (14.33)	1.248	0.231	0.312 (medium)
Shoulder flexion (degrees)	167.38 (12.52)	167 (10.16)	0.178	0.861	0.045 (small)
Non-dominant shoulder					
IR (degrees)	60.69 (10.36)	59.38 (11.12)	0.615	0.547	0.154 (small)
ER (degrees)	78.38 (11.15)	74.63 (9.82)	1.708	0.088 ^a^	0.357 (medium)
Shoulder flexion (degrees)	161.69 (13.31)	162.75 (15.76)	−0.230	0.821	0.058 (small)
Ball release velocity					
Jump shot ball release velocity (m/s^−1^)	21.53 (1.27)	19.98 (2.46)	2.919	0.011 *	0.730 (medium)
Standing shot ball release velocity (m/s^−1^)	22.52 (1.58)	20.97 (3.57)	2.379	0.017 *^a^	0.561 (medium)

*: *p* < 0.05; ^a^: *p* obtained from Wilcoxon Signed-Rank test; ER—External rotation; IR—Internal rotation.

## Data Availability

Data will be available from the corresponding author upon reasonable request.

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
