# Peer review of "Effects of an 11-Week Detraining, Imposed by the COVID-19 Confinement, on Handball Players’ Shoulder Rotator Isokinetic Profile, Shoulder Range of Motion, and Ball Release Velocity"

_medicina, 2023, doi:10.3390/medicina59091548_

Round 1

Reviewer 1 Report

medicina-2528392

Reviewer comments

In the submitted paper, the author(s) examined the possible decrement in isokinetic torque, shoulder range of motion and ball release velocity on handball players due to a 10-week compulsory detraining due to COVID-19 confinement. Sixteen handball players were tested.

There are some concerns that need to be addressed, as listed below:

General Comments

  • Introduction: Elaborate on the measures made to examine the hypothesis of the study. More information is needed about the importance of isokinetic torque evaluation in handball players. In addition, there is no mention about fatigue. Finally, since the results are presented for the dominant and non-dominant arm, a paragraph on the inter-limb difference in the upper arm in handball players in also of importance (see for example: https://doi.org/10.1186/s13102-021-00364-3; https://doi.org/10.1519/JSC.0000000000004422; https://doi.org/10.1080/2331205X.2019.1678221; https://doi.org/10.3390/sports10060086).
  • Methods: Following the first comment, additional rational is needed for the adoption of the selected angular velocities in the isokinetic tests.
  • Discussion: Elaborate about the mechanism responsible for the different detraining results in external and internal shoulder rotations and the effectiveness of the proposed physical conditioning program recommended during the confinement period.
  • Conclusion: Provide more specific guidelines for coaches and practitioners.

Specific comments:

  • L77-89: Provide further anthropometric variables i.e., body mass and height.
  • L77-89: State how many players were left-handed.
  • L97: It is stated that the confinement lasted 10-weeks; however, in L17, it appears that the detraining period lasted 11 weeks. Please, clarify.
  • L99-103: Was the physical fitness program checked for its systematic conduction via a questionnaire?
  • L102: “leg push-ups” = squats?
  • L136-140: Provide in specific which trials were included in W1 and W2.
  • L160-171: Elaborate on the ball release velocity measure: to define the plane a frame rather than a pipe is needed. What was the length of the calibration pipe? How many control/reference points were used on the pipe to calibrate the field of view? What was the reliability to estimate the exact frame of ball release? Also, about the polynomial: what was its degree?
  • L179-182: The mentioned thresholds are interpreting the effect size in the non parametric tests. Provide the respective thresholds for Cohen’s d.
  • L179: It is suggested to provide Cohen’s d in lowercase typo.
  • Results: It is proposed to delete “value” after p.
  • L236: Is this value and unit (22.06°) correct? Such a difference is not presented in neither 3 tables.
  • References: Provide all journal titles as indicated in the Author’s guidelines.

Author Response

Authors´ comment: Thank you for all your constructive and valuable feedbacks. We truly believe that after considering all your comments, the quality of the manuscript has been significantly improved.

  • Introduction: Elaborate on the measures made to examine the hypothesis of the study. More information is needed about the importance of isokinetic torque evaluation in handball players. In addition, there is no mention about fatigue. Finally, since the results are presented for the dominant and non-dominant arm, a paragraph on the inter-limb difference in the upper arm in handball players in also of importance (see for example: https://doi.org/10.1186/s13102-021-00364-3; https://doi.org/10.1519/JSC.0000000000004422; https://doi.org/10.1080/2331205X.2019.1678221; https://doi.org/10.3390/sports10060086).

Authors´ response: Thank you for your constructive feedback. We have included further rationale about these topics in the introduction. In addition, thank you for providing these interesting references. These articles have been cited.

  • Methods: Following the first comment, additional rational is needed for the adoption of the selected angular velocities in the isokinetic tests.

Authors´ response: Thank you for your comment. The requested information has been included in the document, with the addition of a bibliographic reference: Bagordo et al. (2020) - doi:ARTN 12410.3390/sports8090124.

  • Discussion: Elaborate about the mechanism responsible for the different detraining results in external and internal shoulder rotations and the effectiveness of the proposed physical conditioning program recommended during the confinement period.

Authors´ response: Thank you for your recommendation. It has been included.

  • Conclusion: Provide more specific guidelines for coaches and practitioners.

 Authors´ response: Thank you for your constructive feedback. We have provided more specific guidance.

 Specific comments:

  • L77-89: Provide further anthropometric variables i.e., body mass and height.

Authors´ response: Included.

  • L77-89: State how many players were left-handed.

Authors´ response: This information was included in the document.

  • L97: It is stated that the confinement lasted 10-weeks; however, in L17, it appears that the detraining period lasted 11 weeks. Please, clarify.

Authors´ response: Thank you for pointing us this typo. What happened was that the confinement period effectively lasted 10 weeks, however handball training was only resumed 1 week after the confinement, which is why the detraining period lasted 11 weeks.

  • L99-103: Was the physical fitness program checked for its systematic conduction via a questionnaire?

Authors´ response: Thank you for your comment, we fully understand, being a clear limitation. Unfortunately, the physical fitness program was not controlled. But since the program was carried out individually at home during lockdowns, it would also be difficult to be sure that it would have been strictly followed. This aspect has been included at the end of the document as a limitation.

  • L102: “leg push-ups” = squats?

Authors´ response: Yes, it has been clarified.

  • L136-140: Provide in specific which trials were included in W1 and W2.

Authors´ response: Information has been added to the document. The Fatigue Index is defined in accordance with the Biodex recommendations and is calculated by the Biodex software. They consider, as explained, the mechanical work of the first third and last third of the repetitions, respectively. The 1st third of the repetitions are the first four trials, and the last third are the last 4.

  • L160-171: Elaborate on the ball release velocity measure: to define the plane a frame rather than a pipe is needed. What was the length of the calibration pipe? How many control/reference points were used on the pipe to calibrate the field of view?

Authors´ response: Thank you very much for your comment, which will considerably improve the description of the whole procedure. We made changes to the document.

 The calibration was performed using the suggestions presented by the international society of biomechanics and we used the direct linear transformation (DLT). Despite knowing that the minimal number of points required to estimate the DLT-2D parameters is four, we collected seven non-collinear points utilizing a PIPE (2.60m high) stretched over four meters, forming a plane with dimensions of 4x2.60m.

What was the reliability to estimate the exact frame of ball release?

Authors´ response: Thanks for the question. With 210 images the admissible error will be inside 0.0047619 of a second and, from our perspective, for the purpose of this work it is an admissible error, not being necessary to do the reliability procedure. However, if you think it is absolutely necessary, we can perform the process, since we have the images and we can do it.

Regarding the ball velocity, we have made changes to the document, including the following sentence: “The ball velocity was calculated using eight frames, four images before the ball left the player's hand and four images immediately after the ball left the player's hand. Using the projectile motion equations, the horizontal movements had an equation (polynomial of degree 1) while the vertical movements had a quadratic equation (polynomial of degree 2).

  • Also, about the polynomial: what was its degree?

Authors´ response: Thanks for the question. As mentioned in the previous question, we have made changes to the document, including the following sentence: “The ball velocity was calculated using eight frames, four images before the ball left the player's hand and four images immediately after the ball left the player's hand. Using the projectile motion equations, the horizontal movements had an equation (polynomial of degree 1) while the vertical movements had a quadratic equation (polynomial of degree 2).

  • L179-182: The mentioned thresholds are interpreting the effect size in the non-parametric tests. Provide the respective thresholds for Cohen’s d.

Authors´ response: Included.

  • L179: It is suggested to provide Cohen’s d in lowercase typo.

Authors´ response: Modified.

  • Results: It is proposed to delete “value” after p.

Authors´ response: Modified.

  • L236: Is this value and unit (22.06°) correct? Such a difference is not presented in neither 3 tables.

Authors´ response: Thank you for pointing us this typo. It has been corrected. The mean difference between pre and post confinement in the IR ROM is 10.44 degrees.

  • References: Provide all journal titles as indicated in the Author’s guidelines.

Authors´ response: The references were organized in accordance with the journal Author's guidelines. As suggested by the journal, we used a bibliography software package (EndNote), and we have even downloaded the style file made available by the Medicina journal.

In the guidelines for authors, they state that we should include the Abbreviated Journal Name.

“Journal Articles:
1. Author 1, A.B.; Author 2, C.D. Title of the article. Abbreviated Journal Name YearVolume, page range.”

Reviewer 2 Report

Effects of an 11-week detraining, imposed by the COVID-19 confinement, on handball players’ shoulder rotator isokinetic profile, shoulder range of motion, and ball release velocity

First, the reviewer would like to thank the authors for their work and efforts in improving sports science knowledge.

General comments to the authors

Overall, this is a nice study that could have great practical application when integrated with the effects of an 11-week detraining, imposed by the covid-19 confinement, on handball players’ shoulder rotator isokinetic profile, shoulder range of motion, and ball release velocity. The authors are commended on their efforts thus far. The study is well-designed and well-written, with a great original article evaluating the usefulness of the topic. However, I suggest only small corrections and the authors should update the recent references about the detraining; these corrections and studies will allow improving the manuscript.

Abstract

This section is well-designed and well-written.

Introduction and discussion sections

I think that you can add recent references about detraining.

Clemente, F. M., Soylu, Y., Arslan, E., Kilit, B., Garrett, J., van den Hoek, D., ... & Silva, A. F. (2022). Can high-intensity interval training and small-sided games be effective for improving physical fitness after detraining? A parallel study design in youth male soccer players. PeerJ, 10, e13514.

Ruiz-Navarro, J. J., Plaza-Florido, A., Alcantara, J. M., Gay, A., & Arellano, R. (2023). Detraining Effect on Cardiac Autonomic Response to an All-Out Sprint Exercise in Trained Adolescent Swimmers. International Journal of Sports Physiology and Performance, 18(6), 573-578.

Methods section

Is there any G-power analysis? Please add it. If your sample size is small, please add it to the limitations.

Results section

You can add the descriptor of the Cohens d (for example, small, trivial etc.) to Tables 

Discussion section

Overall the discussion is well-written and incorporates relevant literature.

Figures and Tables

This section is well-designed and well-shown.

Author Response

Authors´ comments: Thank you for all your valuable and constructive feedbacks. We truly believe that after consdiering all of your comments, the quality of the manuscript has been significantly improved.

General comments to the authors

 Overall, this is a nice study that could have great practical application when integrated with the effects of an 11-week detraining, imposed by the covid-19 confinement, on handball players’ shoulder rotator isokinetic profile, shoulder range of motion, and ball release velocity. The authors are commended on their efforts thus far. The study is well-designed and well-written, with a great original article evaluating the usefulness of the topic. However, I suggest only small corrections and the authors should update the recent references about the detraining; these corrections and studies will allow improving the manuscript.

Abstract

This section is well-designed and well-written.

 Authors´ response: Thank you for your feedback.

Introduction and discussion sections

I think that you can add recent references about detraining.

Clemente, F. M., Soylu, Y., Arslan, E., Kilit, B., Garrett, J., van den Hoek, D., ... & Silva, A. F. (2022). Can high-intensity interval training and small-sided games be effective for improving physical fitness after detraining? A parallel study design in youth male soccer players. PeerJ, 10, e13514.

Ruiz-Navarro, J. J., Plaza-Florido, A., Alcantara, J. M., Gay, A., & Arellano, R. (2023). Detraining Effect on Cardiac Autonomic Response to an All-Out Sprint Exercise in Trained Adolescent Swimmers. International Journal of Sports Physiology and Performance, 18(6), 573-578.

Authors´ response: Thank you for providing these interesting references. These articles have been cited.

Methods section

Is there any G-power analysis? Please add it. If your sample size is small, please add it to the limitations.

Authors´ response: Included. 

Results section

You can add the descriptor of the Cohens d (for example, small, trivial etc.) to Tables 

 Authors´ response: Included.

Discussion section

Overall, the discussion is well-written and incorporates relevant literature.

 Authors´ response: Thank you for your feedback.

Figures and Tables

This section is well-designed and well-shown.

 Authors´ response: Thank you for your feedback.

Round 2

Reviewer 1 Report

medicina-2528392-R1

Reviewer comments on the resubmission

In the submitted revised version of the text, the author(s) did an exceptional work to address the concerns raised in the initial round of reviewing. However, there are still some topics, as listed below:

Specific comments:

  • L129: Please, clarify if the “leg push-ups” is still a push up but with a support on one leg. If this is the case, was the non-support leg lifted up, straight or bended?
  • L194: Report the sampling frequency in fps (fields per second).
  • L199: …2D-DLT…
  • L202-204: Reliability to estimate the exact frame of ball release: despite the small error (which is proposed to be included in the text and also to report the range of possible error in terms of ball release velocity), the reliability topic in this case concerns the valid identification of the field containing the exact instant of ball release; thus, reporting intra- or inter-rater reliability is of value.
  • L211: SPSS = International Business Machines Corp., Armonk, NY, USA.
  • Tables: No ‘a’ is depicted within the Tables’ cells, despite being mentioned in the Tables’ footnote.
  • References: Not all journals are provided in their abbreviated title (#2, #10-12, #16-18, #28, #34, #38, #44-45).

Author Response

In the submitted revised version of the text, the author(s) did an exceptional work to address the concerns raised in the initial round of reviewing. However, there are still some topics, as listed below:

Authors´response: Thank you again for all your constructive and valuable suggestions. These have inspired us to improve the manuscript´s quality.

Specific comments:

  • L129: Please, clarify if the “leg push-ups” is still a push up but with a support on one leg. If this is the case, was the non-support leg lifted up, straight or bended?

Authors´ response: Thank you for your comment. It has been changed in the document.

  • L194: Report the sampling frequency in fps (fields per second).

Authors´ response: Thank you for your comment. However, the information on sampling frequency in fps is already in the document. In line 193 we have: “The ball velocity was measured using a High-Speed Camera (Casio Exilim EX-FH25) at a 210Hz frequency.”

  • L199: ...2D-DLT...

Authors´ response: Thank you for your recommendation. It has been included.

  • L202-204: Reliability to estimate the exact frame of ball release: despite the small error (which is proposed to be included in the text and also to report the range of possible error in terms of ball release velocity), the reliability topic in this case concerns the valid identification of the field containing the exact instant of ball release; thus, reporting intra- or inter-rater reliability is of value.

Authors´ response: We have made changes in the document.

Considering that the entire digitization process was carried out only by one researcher, the Inter-operator reliability was not assessed. Therefore, only the intra-operator consistency was evaluated (references 1–5). Digitalization’s were performed, four positions before ball release and four positions after ball release of the same execution over five days. The resulting speed calculation was assessed and presented in the table below. Cronbach's alpha, which determines the lower limit of the internal consistency of a group of variables, showed a value of =0.985, which represents very good consistency.

D -Day ; R – Average of 8 

References:

  1. Chernetckiy I, Slipukhina І, Kurylenko N, Mieniailov S, Opachko M. The Application of Tracker Video Analysis for Distance Learning of Physics.
  2. Deltombe T, Detrembleur C, Gruwez G. Comparison of Tracker 2-D video software and Vicon 3-D system in knee and ankle gait kinematic analysis of spastic patients. Annals of Physical and Rehabilitation Medicine. setembro de 2017;60:e51.
  3. Hulka K, Cuberek R, Svoboda Z. Time–motion analysis of basketball players: a reliability assessment of Video Manual Motion Tracker 1.0 software. Journal of Sports Sciences. 2 de janeiro de 2014;32(1):53–9.
  4. Serrano J, Fernandes O. RELIABILITY OF A NEW METHOD TO ANALYSE AND TO QUANTIFY ATHLETES’ DISPLACEMENT. 2011;2.
  5. Serrano J, Shahidian S, Fernandes O. Validation of a Manual Position Tracking Software (TACTO) to Quantify the Football Movements. International Journal of Sports Science. 2014

  • L211: SPSS = International Business Machines Corp., Armonk, NY, USA.

Authors´ response: Thank you for your recommendation. It has been included.

  • Tables: No ‘a’ is depicted within the Tables’ cells, despite being mentioned in the Tables’ footnote.

Authors´ response: Thanks for the careful reading. We have made changes to the tables.

  • References: Not all journals are provided in their abbreviated title (#2, #10-12, #16-18, #28, #34, #38, #44-45).

Authors´ response: Thanks for the careful reading. We have made changes to the references.
